# Prognostic Impact of *EGFR* Amplification and Visceral Pleural Invasion in Early Stage Pulmonary Squamous Cell Carcinomas Patients after Surgical Resection of Primary Tumor

**DOI:** 10.3390/cancers14092174

**Published:** 2022-04-27

**Authors:** Luís Miguel Chinchilla-Tábora, José María Sayagués, Idalia González-Morais, Marta Rodríguez, María Dolores Ludeña

**Affiliations:** Department of Pathology and IBSAL, University Hospital of Salamanca, University of Salamanca, 37007 Salamanca, Spain; lmchinchilla@saludcastillayleon.es (L.M.C.-T.); idalia_dsks@hotmail.com (I.G.-M.); martarodriguez@saludcastillayleon.es (M.R.)

**Keywords:** squamous cell lung carcinoma, *EGFR* amplification, early stage, prognosis

## Abstract

**Simple Summary:**

Lung cancer is the most prevalent malignant neoplasm worldwide. Pulmonary squamous cell carcinoma (SCC) is a histological subtype of non-small cell lung carcinoma with a high rate of morbimortality. The identification of biomarkers with prognosis value is critical in the follow-up and management of SCC patients. Amplification of Epidermal Growth Factor Receptor (*EGFR*) gen on 7p12 is frequently found in pulmonary SCC. We analyzed this alteration with immunohistochemistry, and fluorescence in situ hybridization, along with clinic-pathological variables such as gender, age, pT-stage, lymph node status, pleural invasion, stage, recurrence, metastasis, and overall survival, in a homogeneous population of 108 patients. We found that *EGFR* amplification, exclusively in early stage SCC (pT1–pT2), is an independent adverse prognostic factor for overall survival and combined with visceral pleural invasion, allows the identification of three groups of pulmonary SCC patients with significantly different outcome, which could be established at diagnosis.

**Abstract:**

Over the last few decades, an increasing amount of information has been accumulated on biomarkers in non-small cell lung cancer (NSCLC). Despite these advances, most biomarkers have been identified in the adenocarcinoma histological subtype (AC). However, the application of molecular-targeted therapies in the prognosis and treatment of SCC in the clinical setting is very limited, becoming one of the main focus areas in research. Here, we prospectively analyzed the frequency of numerical/structural abnormalities of chromosomes 5, 7, 8, 9, 13 and 22 with FISH in 48 pulmonary SCC patients. From a total of 12 probes, only abnormalities of the 7p12 and 22q12 chromosomal regions were identified as unique genetic variables associated with the prognosis of the disease. The study for these two chromosomal regions was extended to 108 patients with SCC. Overall, chromosome losses were observed more frequently than chromosome gains, i.e., 61% versus 19% of all the chromosome abnormalities detected. The highest levels of genetic amplification were detected for the 5p15.2, 7p12, 8q24 and 22q11 chromosome bands, of which several genes are potentially involved in the pathogenesis of SCC, among others, include the *EGFR* gene at chromosome 7p12. Patients who displayed *EGFR* amplification (n = 13; 12%) were mostly older than 65 years (*p* = 0.07) and exclusively patients in early T-primary tumor stage (pT1–pT2; *p* = 0.03) with a significantly shortened overall survival (OS) (*p* ≤ 0.001). Regarding prognosis, the clinical, biological, and histopathologic characteristics of the disease that displayed a significant adverse influence on OS in the univariate analysis included patients older than 65 years (*p* = 0.02), the presence of lymph node involvement (*p* = 0.005), metastasis (*p* = 0.01) and, visceral pleural invasion (VPI) at diagnosis (*p* = 0.04). EGFR amplification also conferred an adverse impact on patient OS in the whole series (*p* = 0.02) and especially in patients in early stages (pT1–pT2; *p* = 0.01). A multivariate analysis of the prognostic factors for OS showed that the most informative combination of independent variables to predict an adverse outcome was the presence of VPI and/or *EGFR* amplification (*p* < 0.001). Based on these two variables, a scoring system was built to stratify patients into low- (no adverse features: score 0; n = 69), intermediate- (one adverse feature: score 1; n = 29) and high-risk (two adverse features: score 2; n = 5) groups, with significantly different (*p* = 0.001) OS rates at 50 months, which were as following: 32%, 28% and 0%, respectively. In the present study, we show that the presence of a high level of 7p12 (*EGFR)* amplification, exclusively detected in early stage SCC (pT1–pT2), is an independent adverse prognostic factor for OS. The identification of the *EGFR* gene copy number using FISH techniques may provide a more accurate diagnosis of high-risk populations after the complete resection of the primary tumor. When combined with VPI, three groups of pulmonary SCC were clearly identified that show the extent of the disease. This is of such importance that further prospective studies are necessary in larger series of SCC patients to be classified at the time of diagnosis. This could be achieved with the combined assessment of 7p12 amplification and VPI in primary tumor samples.

## 1. Introduction

Lung cancer is the most common disease in human beings, with the highest morbimortality rates, reaching the score of 18%, representing 1.8 million deaths. Non-small cell lung carcinoma (NSCLC) constitutes 85% of the total lung cancer cases. The pulmonary squamous cell carcinoma (SCC) subtype corresponds to 30% of NSCLC [1].

Over the last few decades, an increasing amount of information has been compiled on biomarkers in this type of patient [2]. Despite these advances, most biomarkers have been identified in the adenocarcinoma histological subtype (AC). In this regard, although the *KRAS* mutation is the most common, epidermal growth factor receptor (*EGFR*) mutations and anaplastic lymphoma kinase (*ALK*) rearrangements deserve special attention for their clinical significance.

AC has more effective (more than 1 in 3 patients) molecularly targeted therapies [3], whereas the treatment and prognosis of SCC is still limited, currently being one of the main fields of research.

Pulmonary SCC subsets have been described by The Cancer Genome Atlas (TCGA), to try to develop new effective targeted therapies in order to tailor to each patient individually [4,5].

Prognosis gene signatures for risk stratification confine the selection to patients who would require adjuvant therapies, especially in patients in the early stages of the disease [6]. In this sense, *EGFR* represents the most frequently mutated driver gene in lung cancer. *EGFR* stimulate several intracellular signaling pathways in NSCLC. AC and SCC are NSCLC subtypes with distinct clinic-biological and molecular features [7]. In comparison with lung AC, *EGFR* mutations are relatively rare in the SCC of the lung, with a reported mean prevalence of 7% [8]. In contrast, amplification at the level of this gene is much more frequent, represents an early event in patients with SCC and could be associated with prognosis and/or treatment of the disease, as indicated in the literature [9]. Recently, Hirsch et al. showed that SCC patients with both *EGFR* amplification by fluorescence in situ hybridization (FISH) and *EGFR* overexpression by immunohistochemistry (IHC) improved in terms of overall survival (OS) when Cetuximab was added to chemotherapy [10]. Currently, *EGFR* amplification is being used as an inclusion criterion in three clinical trials for squamous cell lung carcinoma with *EGFR* tyrosine kinase inhibitors (TKIs). In addition, Cappuzzo et al. also found that *EGFR*-TKIs had a better curative effect in patients with NSCLC carrying *EGFR* amplifications [11]. Kato et al. examined the genomic landscape of *EGFR* amplification by interrogating blood-derived cfDNA from 28,584 patients with diverse malignancies using NGS, observing high response rates among patients who harbored *EGFR* amplification and anti-*EGFR*–based therapies [12].

Other studies have shown that the *EGFR* gene copy number or amplification detected using FISH could be a good biomarker for predicting the prognostic in patients with NSCLC. Jia et al. showed that NSCLC patients with lymph node metastasis presented higher *EGFR* gene amplification rates [13]. *EGFR* gene amplification is related to adverse clinical outcomes in other squamous cell carcinomas, such as uterine cervix [14], oral cavity [15], head and neck, and esophagus [16,17]. These results indicate that *EGFR* amplification might be closely associated with invasion and metastasis, making it a novel prognostic biomarker and a therapeutic target.

In the present study, we investigated the prognostic value of structural/numerical abnormalities detected with interphase FISH involving the most frequent chromosomes involving 108 cases of pulmonary SCC with a long median follow-up.

## 2. Materials and Methods

### 2.1. Patients and Samples

In this study, we retrospectively analyzed surgical specimens from 48 consecutive patients diagnosed with pulmonary SCC between 1999 and 2010. Those clinic-pathological features and chromosomal regions that were found to have a significant influence on the prognosis of the disease were extended to the study with 108 patients (105 males and 3 females; with a median age of 66 years, ranging from 27 to 83 years). Every patient provided their informed consent before the beginning of this study. All subjects underwent a surgical resection of primary tumor in our institution, and they were diagnosed and classified according to the WHO criteria [18,19], prior to any treatment. According to the tumor stage, 85 cases were classified as early stages (pT1–pT2 tumors) and 23 as late stages (pT3–pT4 tumors).

We completely confirmed the histopathological features and grade in a second evaluation that was conducted by an expert and independent pathologist. The study closed with a median follow-up time of 47 months (range: 26–58 months). The study was authorized by the local ethics committee of the University Hospital of Salamanca (Salamanca, Spain) and we also previously obtained the informed consent of each patient under study. Table 1, Table 2 and Table 3 provides the clinical, laboratory, genetic, and follow-up data of patients.

### 2.2. Tissue Microarray (TMA)

We extracted a 0.6 mm diameter core biopsy fragment from the FFPE block of tissue, containing biopsies of the primary tumors from all patients (n = 108) using a tissue microarray instrument. Tissue cores from each specimen were embedded together with paraffin after a pathologist selected the histology sections. All tumor samples were included in triplicate in TMAs and analyzed cores of normal tissue of the same patients were used as controls. Three squamous cell lung cancer TMAs were completed using a Beecher Manual Tissue Microarrayer MTA-1.

### 2.3. Immunohistochemistry

Formalin fixed paraffin embedded (FFPE) TMA sections (4-μm-thick) were deparaffinized in xylene and rehydrated. The antigen retrieval was performed using a heater in 0.01 M citrate buffer and pH 6.0. Inactivation of endogenous peroxidase was fulfilled with 0.3% H_2_O_2_ in methanol for 15 min.

Antibodies and reagents were included automatically using the Leica BOND-III processor (Leica Biosystems, Newcastle, CA, USA), using the standard protocols and recommendations of the manufacturers. Sections were counterstained with hematoxylin, dehydrated and mounted. Clones, dilutions, manufacturers and interpretations of the primary antibodies used are specified in Appendix A. It is worth noting EGFR IHC expression criteria used: (0/+), no membranous stain or low expression in tumour cells; (++), 1–35% of tumour cells with moderate membranous stain; and (+++), <35% of tumour cells with overexpressed membranous stain.

### 2.4. FISH Analysis

The FISH studies for the 5p15.2, 5q31, 7q31, 8p11.1, 8q24, 9p21, 9p11.1, 13q14 and 13q34 chromosomal regions were performed on 48 SCC patients on a single TMA slide, whereas studies for chromosomal regions 7p12, 7p11.1 and 22q12 were performed on 108 cases with SCC in three different TMAs.

TMA sections in slides were dried overnight at 60 °C and deparaffinized in xylene, followed by a 13 min water bath at 80 °C into Vysis Paraffin Pretreatment Reagent (Vysis; Downers Grove, IL, USA). The slides were rinsed with deionized water and immediately treated with protease solution (250 mg pepsin + 62.5 mL 0.2 N HCl, pH 1.0) for 13 min at 37 °C in a water bath. The next step was to rinse them thoroughly again with water and leaving them to air dry, and 20 µL of probe was added to each slide. A set of 12 different probes (Vysis Inc., Downers Grove, IL, USA) specific for those chromosomes and chromosomal regions most frequently gained/amplified and deleted in SCC [20], were systematically used in double staining with the Spectrum Orange (SO) and Spectrum Green (SG) fluorochromes as follows: for chromosome 5, the LSI D5S721 (5p15.2) (SG)/EGR1 (5q31) (SO) Dual Color probe was used; for chromosome 7, the LSI D7S486 (7q31) (SO), LSI *EGFR* (SO)/CEP 7 (SG) Multi-color probe was employed; for chromosome 8, the CEP 8 (SG)/*MYC* (8q24) (SO) was used; for chromosome 9, the CEP 9 (SG)/*CDKN2A* (9p21) (SO) was used; for chromosome 13, the LSI *RB1* 13q14 (SO)/LSI 13q34 *LAMP1* (SG) probe combination was employed, and; for chromosome 22, the LSI *EWRS1* (22q11.2) probe was used (Appendix A).

We sealed each slide with coverslips using rubber cement. Then, the slides were denatured at 75 °C for 5 min and overnight hybridization at 37 °C in a Hybrite thermocycler (Vysis) was required. Slides were subjected to a subsequent rinsing which consists of in 2 X SSC with 0.3% NP-40 at 73 °C for 1 min in a waterbath. Slides were allowed to air dry in the dark before being counterstained with 20 µL of DAPI (Sigma, St. Louis, MO, USA). We used the commercial antifading agent Vectashield (Vector Laboratories, Burlingame, CA, USA) and a BX60 fluorescence microscope (Olympus, Hamburg, Germany) equipped with a 100 × oil objective to quantify the number of hybridization spots per nuclei of ≥200 cells per sample. *EGFR* gene amplification was defined as an *EGFR*/CEP7 ratio ≥ 2, in accordance with the manufacturer’s recommendations (Vysis; Downers Grove, IL, USA). In addition, we included the term “high levels” of amplification when it was found in the form clusters of hybridization signals.For each specimen, it was possible to quantify at least 200 tumor nuclei. Simultaneously, two experienced pathologists interpreted the results, and they were in full agreement.

### 2.5. Statistical Analyses

To evaluate the quality of statistical data for the group differentiations, we used Student’s t and Mann–Whitney U tests. The former was used for normally distributed continuous variables and the latter was used for non-normally distributed continuous variables. SPSS v.22 (IBM Corp., Armonk, NY, USA) was used to calculate continuous variables, as well as the mean, and the standard deviation (SD). Frequencies and percentages were indicated as our dichotomous variables using X^2^ test. The statistical significance of the differences between survival curves were determined with the one-sided long-rank test, while overall survival (OS) curves were patterned according to the Kaplan–Meier method. Multivariate analyses of prognostic factors for OS were identified by multivariate step-wise Cox regression, considering only those variables that display a meaningful association with OS in the univariate analysis. A *p*-value of < 0.05 (or, if applicable, Pearson-corrected *p*) was considered statistically significant.

## 3. Results

### 3.1. Chromosomal Alterations in SCC

All patients with pulmonary SCC included in this study had numerical abnormalities for at least 1 of the 12 chromosomal regions analyzed (Figure 1). Overall, chromosome losses were observed more frequently than chromosome gains, i.e., 61% versus 19% of all numerical chromosome abnormalities detected. However, once individual chromosomes were considered, 9p21 and 7q31 chromosomal regions were found to be the most frequently altered (90% and 96%, respectively). In addition, the highest levels of genetic amplification were detected for the 5p15.2, 7p12, 8q24 and 22q11 chromosome bands, where several genes were potentially involved in the pathogenesis of SCC. Among others, these include the *EGFR* gene at chromosome 7p12 and the *CMYC* gene at chromosome 8q24. The most commonly amplified single region (10/48; 21%) corresponded to a region located at chromosome 5p15.2, where loci D5S23 and D5S721 are found (Figure 1).

### 3.2. Prognostic Effect of Chromosomal Changes and Other Disease Features in SCC Patients

From a prognostic point of view, significant association was found between the clinical and biological features of the disease and the OS of SCC patients, as expected at the advanced pathological stage (pT3–pT4; *p* = 0.001), microscopically positive resection margins (R1; *p* = 0.003), local recurrences (*p* = 0.05), and the presence of metastases (M1; *p* = 0.05). By contrast, no significant differences (NS) were found between OS on SCC cases regarding patient gender, age, lymph node status, visceral pleural invasion (VPI), and AJCC stage (Table 1). From a genetic point of view, the presence of gains/amplifications of 7p12 (*p* = 0.02) meant that it was identified as the sole chromosomal region with a significantly inferior outcome (Table 2).

### 3.3. Clinical, Histopathological, and Biological Disease Characteristics of SCC Patients According to the Number of Copies of EGFR Gene Detected by FISH Techniques

Those clinic-pathological features and chromosomal region of prognosis value were extended to the study with 108 patients. Table 3 shows the clinical, histopathologic, and biologic disease characteristics of the 108 SCC patients analyzed according to the number of copies of the *EGFR* gene detected per cell (Figure 2). As this table illustrates, patients who displayed *EGFR* amplification (n = 13; 12%) were mostly older than 65 years (*p* = 0.07) and exclusively patients in early T-primary tumor stage (pT1–pT2; *p* = 0.03) with a significantly shortened OS (*p* ≤ 001). Interestingly, most of the cases with *EGFR* amplification were located on the right lung (n = 10, 77%; *p* = 0.05).

Information on smoking habits was available for 49 patients of the whole series and 46 (94% of cases) were related to smoking. The four cases with *EGFR* amplification detected in this cohort of patients were found in the group of active smokers.

Regarding prognosis, the clinical, biological, and histopathologic characteristics of the disease that displayed a significant adverse influence on OS in the univariate analysis included patients older than 65 years (*p* = 0.02), the presence of lymph node involvement (*p* = 0.005), metastasis (*p* = 0.01) and visceral pleural invasion (VPI) at diagnosis (*p* = 0.04) (Figure 3). As shown in Figure 4, *EGFR* amplification also conferred an adverse impact on patient OS in the whole series (*p* = 0.02) exclusively in the early stages (pT1–pT2; *p* = 0.01). A multivariate analysis of the prognostic factors for OS showed that the most informative combination of independent variables to predict an adverse outcome was the presence of VPI and/or *EGFR* amplification (*p* < 0.001). Based on these two variables, a scoring system was built to stratify patients into low- (no adverse features: score 0; n = 69), intermediate- (one adverse feature: score 1; n = 29) and high-risk (two adverse features: score 2; n = 5) groups with significantly different (*p* = 0.001) OS rates at 50 months, of 32%, 28% and 0%, respectively, (Figure 5).

### 3.4. EGFR Protein Expression and Copy Number

There was a significant concordance between *EGFR* IHC positivity and *EGFR* gene copy number gains determined by FISH (*p* < 0.001). In 53 cases (49%) of the 108 patients analyzed, *EGFR* membranous immunoreactivity was observed with IHC, of which 41 (78%) were negative (0) or showed a low (+) to moderate (++) level of *EGFR* IHC expression and were disomic (69%) or showed a high proportion of polysomies (31%) with the FISH analysis, whereas all tumors with *EGFR* IHC overexpression (+++) 12 (22%) showed amplification (100%) by FISH. There was only one case with *EGFR* amplification on FISH analyses that was negative by IHC. *EGFR* IHC positivity was not shown to have a significant influence on disease prognosis, with the univariate analysis (*p* = 0.125).

## 4. Discussion

Based on the analysis of 48 SCC patients, we observed that all cases showed numerical/structural abnormalities for one or more chromosomes in SCC, as reported in previous studies [20,21]. Our results show that the overall incidence of chromosome losses is higher than that of chromosome gains. However, it should be noted that highest levels of genetic amplification were detected at 5p15.2, 7p12, 8q24 and 22q11. Several genes which are potentially involved in the pathogenesis of pulmonary SCC are localized in these four chromosomal regions. Among others, these include the *TERT* and *CLPTM1L* genes on chromosome 5p15, the *EGFR* gene at 7p12, *MYC* at 8q24 chromosomal region and the *CRKL*, *PRKM1* and *MAPK1* genes in chromosome 22q11 [22,23]. It is noteworthy that in our large series of patients we found high levels of *EGFR* amplification in the form of a cluster of spots, exclusively in patients with early stage SCC tumors (pT1–pT2), while *EGFR* gains in patients with advanced-stage SCC tumors (pT3–pT4) were detected in the context of an hyperdiploid karyotype, in the form of 7p12 polysomies. According to our previous findings using FISH, a gain of 7p12 also was observed with significantly greater frequency among patients with early stage SCC tumors (II + IIIA vs. IIIB + IV) [24]. In addition, Kang et al. reported that *EGFR* is expressed in all histotypes of lung carcinoma, but they also found that the SCC tumors usually show higher levels of expression than adenocarcinoma (AC) histology [24]. In agreement with these observations, several authors have previously found that *EGFR* overexpression is habitual in NSCLC, and is found almost exclusively in SCC tumors [9,25]. These results and our findings suggest that high levels of *EGFR* amplification depend on the histological subtype and clinical stage.

The literature related to specific genetic abnormalities in SCC reveals disagreement among authors. As an example, the incidence of *EGFR* gain/amplification ranges between 82% and 4% of all SCC patients [24,26]. Some of these dualities are due to the small number of cases analyzed in a lot of studies [24], the specimen provenance, and the variability in the methods used for the specific genetic abnormalities [25,27,28,29,30]. Molecular approaches such as array comparative genomic hybridization (aCGH) or single nucleotide polymorphism arrays (aSNPs) have several limitations in the detection of losses or gains in genetic material [30,31]. The clonal tumor heterogeneity and genetic diversity in the same tumor sample, particularly when tumor cell clones are present at relatively low frequencies, do not show detailed information when these probes are used. Such difficulties are solved using FISH techniques with an adequate combination of probes, to obtain detailed data about the genetic heterogeneity of tumors at the single-cell level [32,33,34].

From the clinical point of view, *EGFR* gene amplifications have been related to adverse clinical outcomes in SCC from different sites, including esophagus SCC, uterine cervix SCC, and oral SCC, among others. Jiang D. et al. found that *EGFR* amplification and overexpression in esophageal SCC were both significantly correlated with lymph node metastasis with statistically significant results (*p* = 0.04) and (*p* = 0.02), respectively [35]. The results obtained by K Iida et al. showed that *EGFR* amplification is rarely correlated with shorter overall survival (OS) (*p* = 0.001) in uterine cervix SCC. They also found that *EGFR* gene amplification was an independent prognostic factor for OS (*p* = 0.01) in the multivariate analysis [14]. *EGFR* amplification for oral SCC was analyzed by Huei-Tzu Chien et al. They found that *EGFR* copy number alterations (CNAs) are highly associated with clinical stage, tumor differentiation, lymph node metastasis, and a negative effect on oral SCC tumor progression [36].

In agreement with our findings, it is well known that SCC diagnosed in elderly patients (≥80 years) presents a poor prognosis compared with those patients diagnosed ≤ 50-year-old [37]. Faruk Tas et al. [38], demonstrated that age is one of the major prognostic factors affecting survival in lung cancer patients, showing that the median survival time of elderly patients was statistically significantly lower than in younger patients in both univariate (*p* = 0.009) and multivariate (*p* = 0.023) analyses.

It is noteworthy that in most of the cases with *EGFR* amplification, the tumor was located on the right lung (10/13; 77% of cases *EGFR* amplified), in line with previous studies. McWilliam et al. observed that the laterality of NSCLC may be related to differences in the prognosis of the disease. They found in a multivariate analysis, laterality becomes highly significant (*p* < 0.01) and that right-sided tumors have a worse OS than left-sided tumors (15 vs. 18 months; respectively) [39].

Several studies have shown that the *EGFR* gene copy number or amplifications detected by FISH could be a good biomarker for predicting treatment response to *EGFR*-TKIs in patients with advanced NSCLC [40,41,42]. Recently, two phase III trials compared Erlotinib or Gefitinib vs. placebo in the second or third line setting in unselected patients with advanced NSCLC. Both studies showed that the subset of patients with high levels of *EGFR* gene copy number may benefit from *EGFR*-TKI therapy [43,44]. Zhang et al. performed a meta-analysis that involved 17 studies with a total of 2047 patients and analyzed the relationship between the number of copies of the *EGFR* gene and the response to treatment with *EGFR*-TKI in patients with advanced NSCLC. They revealed that patients with high levels of *EGFR* amplification showed increased OS and PFS in patients with advanced NSCLC, receiving *EGFR*-TKI [45]. Similarly, other studies have been carried with the use of anti-EGFR therapies for gastric cancer patients with *EGFR* amplification and/or patients who are EGFR positive with IHC after the surgical resection of primary tumor [46], showing that EGFR positivity is an independent favorable prognostic factor, especially in stage III disease [46]. Based on the poor prognosis conferred by high levels of *EGFR* amplification in patients with early stage SCC tumors (pT1–pT2), and given its high frequency, clinical trials aimed at this purpose deserve further investigation.

In addition to *EGFR* gene amplification, in this study, we also found an association between visceral pleural invasion (VPI, detected in 20% cases), and disease outcome, in line with previous observations. Shimizu et al. [47]), identified VPI in 31 of 138 resected patients (22% of the cases) and they indicated that VPI is an independent predictor of poor prognosis with or without lymph node involvement. The prognostic implications of VPI in resected NSCLC have been well established. In the meta-analysis developed by Ting Wang et al., they investigated the prognostic role of PL0, PL1 (22% of cases) and PL2 (8%) on resected NSCLC patients and found that patients with PL1 and PL2 had worse OS, 5-year survival rate and relapse free survival (RFS) than those with PL0. Moreover, patients with PL2 have even worse OS, 5-year survival rate and RFS than those with PL1 [48]. In line with their results, our findings also demonstrate that VPI adversely impacts the prognosis of resected SCC patients differentially along with the degree of pleural invasion.

A multivariate analysis of prognostic factors for OS showed the independent prognostic value of the two variables, high levels of *EGFR* amplification detected by FISH and VPI (PL1 or PL2); consequently, the coexistence of both adverse features was associated with a significantly reduced OS vs. cases which showed neither of these adverse features (OS at 4 years of 0% versus 32%, respectively). Even though an association has been reported between different chromosomal abnormalities and the prognosis of pulmonary SCC [9,49,50], this is the first report in which the independent prognostic value of amplification of *EGFR* restricted to early stage and VPI is described in a large series of patients with pulmonary SCC.

Finally, the high percentage of males in our patient cohort is striking. The incidence of SCC tumors is lower among women than men, ranging from 10% to 55%; respectively [51]. However, it is strongly correlated with a history of tobacco smoking, regardless of the patient’s gender. Overall, more males develop lung cancer simply due to the fact that males are more likely to smoke [52]. In addition to tobacco, it is believed that genetic factors and biological susceptibility between genders may explain the disparity [51].

In this study, we show that the presence of high levels of 7p12 (*EGFR)* amplification, exclusively detected in early stage SCC (pT1–pT2), is an independent adverse prognostic factor for OS. The identification of the *EGFR* gene copy number by FISH techniques may provide a more accurate diagnosis of high-risk populations after the complete resection of a tumor. By combining this with VPI, we were able to identify three groups of pulmonary SCC patients with significantly different outcomes, which could be predicted at diagnosis. Further prospective studies are required in a larger number of SCC patients to evaluate the prognostic value of the combined assessment of 7p12 amplification and VPI in primary tumor samples at diagnosis and the precise role of the gained genes.

## Figures and Tables

**Figure 1 cancers-14-02174-f001:**
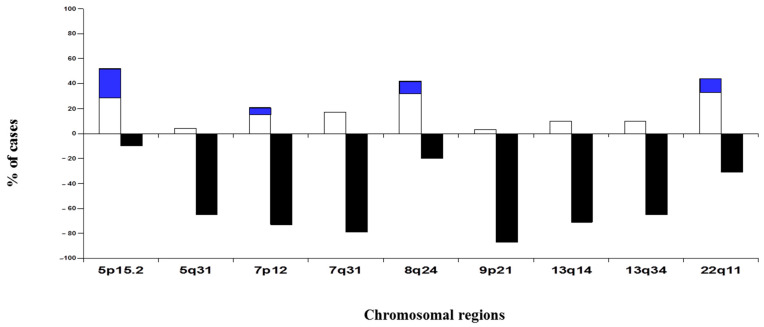
Chromosomal abnormalities detected in pulmonary squamous cell carcinomas (SCC) patients by FISH techniques (n = 48). This summary plot illustrates the frequency of copy number gains (plotted in white above zero values in the *x*-axis) and losses (plotted in black below zero values in the *x*-axis) identified in patients with SCC of the lung. The blue colour indicates the accumulated frequency of the amplification of the chromosomal region analysed, being able to discriminate the polysomies of the amplification of the gene.

**Figure 2 cancers-14-02174-f002:**
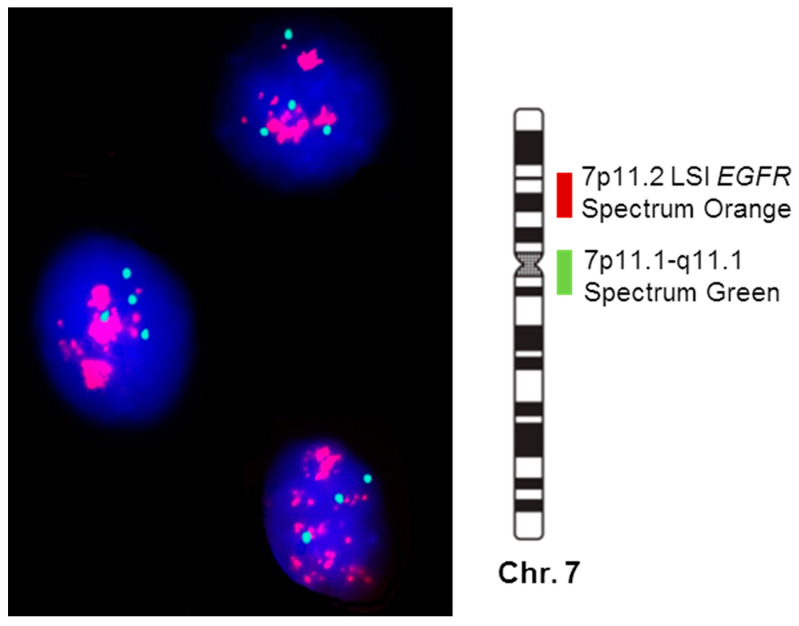
Interphase nuclei with the amplified *EGFR* gene detected by FISH techniques from a biopsy sample of a patient with squamous cell carcinoma of the lung (probes for identifying chromosome 7 centromere (7p11; green spots) and *EGFR* gene (7p11.2; red spots)).

**Figure 3 cancers-14-02174-f003:**
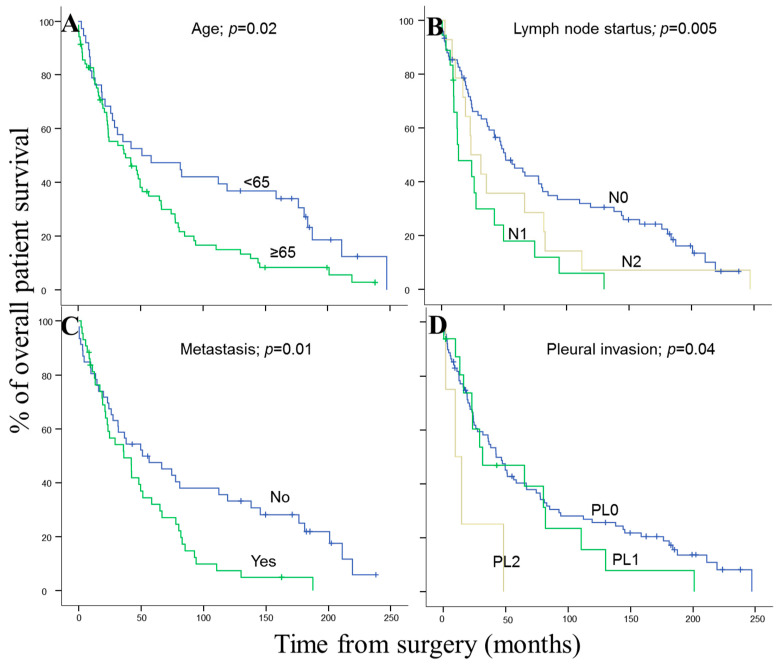
Clinic, histopathologic and biologic-disease characteristics of pulmonary SCC patients which showed a significant impact on overall survival in the univariate analysis: (**A**) Age, (**B**) lymph node status, (**C**) presence of distant metastasis and (**D**) pleural invasion. Survival information was available for 103 cases.

**Figure 4 cancers-14-02174-f004:**
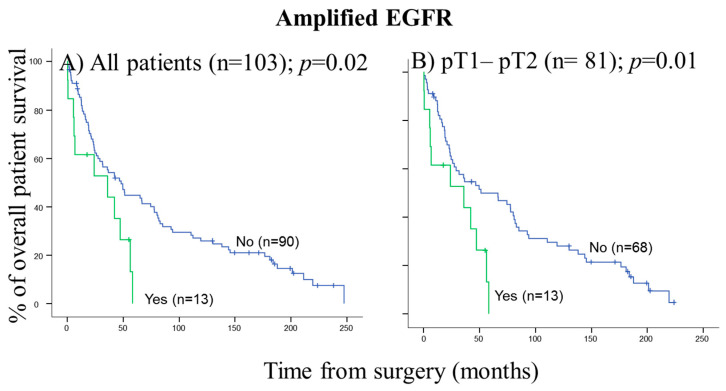
*EGFR* amplification detected in primary tumors from patients with pulmonary SCC by FISH techniques which showed a significant impact on overall survival in the univariate analysis: (**A**) all patients and (**B**) only in early stages: pT1–pT2 patients. Survival information was available for 103 cases.

**Figure 5 cancers-14-02174-f005:**
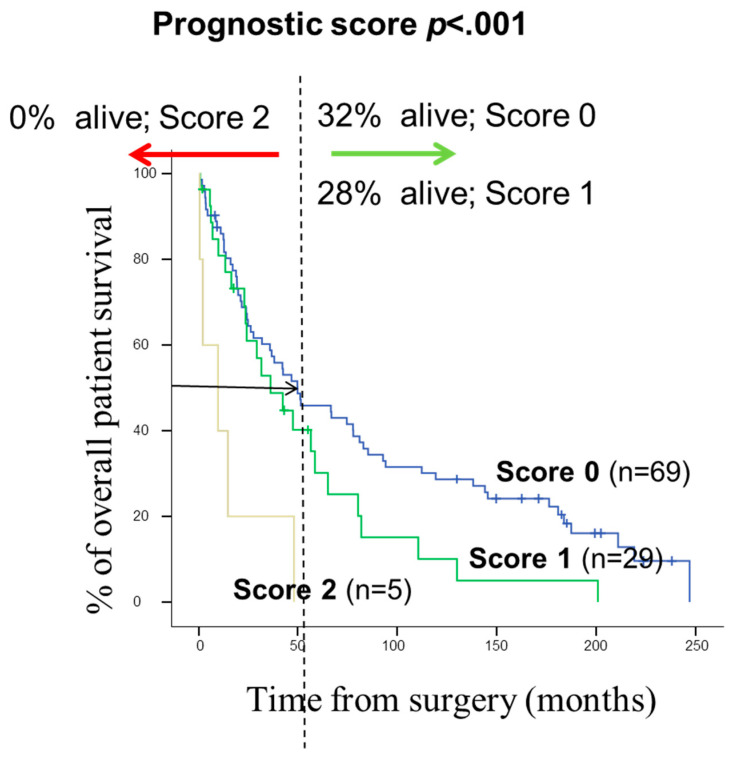
Prognostic score proposed for pulmonary SCC patients, based on the two most informative prognostic factors: pleural invasion, and *EGFR* status; *p* < 0.001. Survival information was available from 103 cases.

**Table 1 cancers-14-02174-t001:** Prognostic impact of clinic-pathological characteristics on overall survival (OS) in 48 patients with pulmonary squamous cell carcinoma (SCC).

Variable	N° of Cases (%)	Median OS, (Months)	*p*
**Gender**			
Male	47 (98)	42	NS
Female	1 (2)	18	
**Age**			
<65	16 (33)	26	NS
≥65	32 (67)	42	
**T-primary tumor stage**			
pT1–pT2	36 (75)	59	0.001
pT3–pT4	12 (25)	22	
**Lymph node status**			
pN0	35 (73)	42	NS
pN1	8 (17)	25	
pN2	5 (10)	39	
pN3	0 (0)	0	
**Pleural invasion**			
PL0	39	42	NS
PL1	6	23	
PL2	3	15	
**AJCC stage**			
I	24	50	
II	14	26	NS
III	9	22	
IV	1	39	
**Type of surgery**			
R0	45	48	0.003
R1	3	10	
**Local recurrence**			
No	19	67	0.05
Yes	29	36	
**Metastasis**			
No	26	52	0.05
Yes	22	39	

AJCC indicates American Joint Committee on Cancer (based on 8th ed. of the AJCC); NS: statistically nonsignificant (*p* > 0.05); pT1: Tumor ≤ 3 cm in greatest dimension, surrounded by lung or visceral pleura, without bronchoscopic evidence of invasion more proximal than the lobar bronchus (i.e., not in the main bronchus); pT2: Tumor > 3 cm but ≤5 cm; pT3: Tumor > 5 cm but ≤7 cm in greatest dimension; or directly invading any of the following: parietal pleura (PL3), chest wall (including superior sulcus tumors), phrenic nerve, parietal pericardium; or separate tumor nodule(s) in the same lobe as the primary; pT4: Tumor > 7 cm in greatest dimension; or tumor of any size invading one or more of the following: diaphragm, mediastinum, heart, great vessels, trachea, recurrent laryngeal nerve, esophagus, vertebral body or carina; or separate tumor nodule(s) in an ipsilateral lobe different from that of the primary; pN0: No regional lymph node metastasis; pN1: Metastasis in ipsilateral peribronchial and/or ipsilateral hilar lymph nodes, and intrapulmonary nodes, including involvement by direct extension; pN2: Metastasis in ipsilateral mediastinal and/or subcarinal lymph node(s); pN3: Metastasis in contralateral mediastinal, contralateral hilar, ipsilateral or contralateral scalene, or supraclavicular lymph node(s); PL0: tumor do not penetrate the visceral pleural elastic layer; PL1: visceral pleural invasion is present when tumor penetrates beyond the elastic layer of visceral pleura; PL2: visceral pleural invasion with tumor extension to the visceral pleural surface; R0: microscopically negative tumor infiltration of resection margins; R1: microscopically positive tumor infiltration of resection margins.

**Table 2 cancers-14-02174-t002:** Frequency of distinct numerical and structural chromosomal abnormalities identified for each chromosome and chromosome region analyzed in 48 pulmonary squamous cell carcinoma (SCC) patients detected using FISH and their impact on overall survival.

Chromosomal Region	Cases (%)	Median OS (Months)	*p*
**Chr. 5p15.2**			
Loss	7 (10)	81	
Normal	18 (38)	48	
Gain	23 (52)	42	NS *
Polysomies	13 (56)	42	
Amplified	10 (44)	42	
**Chr. 5q31**			
Loss	31 (65)	31	
Normal	15 (31)	50	NS *
Gain *	2 (4)	36	
**Chr. 7p12**			
Loss	35 (73)	24	
Normal	3 (6)	42	
Gain	10 (21)	36	0.02
Polysomies	5 (50)	38	
Amplified	5 (50)	32	
**Chr. 7q31**			
Loss	38 (79)	31	
Normal	2 (4)	49	NS *
Gain *	8 (17)	58	
**Chr. 8q24**			
Loss	10 (20)	25	
Normal	18 (38)	31	
Gain	20 (42)	42	NS *
Polysomies	16 (80)	42	
Amplified	4 (20)	42	
**Chr. 9p21.1**			
Loss	42 (87)	36	
Normal	5 (10)	36	NS *
Gain *	1 (3)	42	
**Chr. 13q14**			
Loss	34 (71)	49	
Normal	9 (19)	55	NS *
Gain *	5 (10)	49	
**Chr. 13q34**			
Loss	31 (65)	42	
Normal	12 (25)	55	NS *
Gain *	5 (10)	49	
**Chr. 22q11**			
Loss	15 (31)	74	
Normal	12 (25)	31	
Gain	21 (44)	26	0.09
Polysomies	17 (80)	26	
Amplified	4 (20)	14	

* NS: statistically nonsignificant (*p* > 0.05); OS: Overall survival. The gains corresponded to polysomies of the chromosomal region studied.

**Table 3 cancers-14-02174-t003:** Clinical, histopathological, and biological disease characteristics of squamous cell carcinomas of the lung (SCC), according to the number of copies of EGFR gene detected by FISH techniques (n = 108).

Characteristic	No. Patients [n (%)] Amplified *EGFR*			Total of Cases (n = 108)
	Yes (n = 13)	No (n = 95)	*p*	
**Gender**				
Male	12 (92)	93 (97)	NS	105 (97)
Female	1 (8)	2 (3)		3 (3)
**Age (years)**				
<65	2 (15)	35 (37)	0.07	37 (34)
>65	11 (85)	60 (63)		71 (66)
**Tumor location**				
Right upper lobe	5 (38)	28 (30)	NS	33 (31)
Middle lobe	1 (8)	5 (5)		6 (5)
Right lower lobe	4 (31)	15 (16)		19 (18)
Left upper lobe	2 (15)	20 (21)		22 (20)
Left lower lobe	1 (8)	27 (28)		28 (26)
**Tumor laterality**				
Right	10 (77)	48 (51)	0.05	58 (54)
Left	3 (33)	47 (49)		50 (46)
**T-primary tumor stage**				
pT1–pT2	13 (100)	72 (76)	0.03	85 (79)
pT3–pT4	0 (0)	23 (24)		23 (21)
**Lymph node status**				
pN0	10 (77)	66 (69)	NS	76 (70)
pN1	3 (23)	15 (16)		18 (17)
pN2	0 (0)	14 (15)		14 (13)
pN3	0 (0)	0 (0)		0 (0)
**Pleural invasion**				
pL0	12 (92)	74 (89)	NS	86 (80)
pL1	1 (8)	17 (18)		18 (17)
pL2	0 (0)	4 (4)		4 (3)
**AJCC stage**				
I	7 (53)	45 (47)	NS	52 (48)
II	6 (47)	29 (30)		35 (32)
III	0 (0)	19 (20)		19 (18)
IV	0 (0)	2 (3)		2 (2)
**Local recurrence**				
No	3 (33)	27 (35)	NS	30 (34)
Yes	6 (76)	51 (65)		57 (66)
**Metastasis**				
No	3 (37)	39 (50)	NS	42 (49)
Yes	5 (63)	39 (50)		44 (51)
**Exitus**				
No	2 (15)	14 (15)	NS	16 (15)
Yes	11 (85)	78 (85)		89 (85)
**OS (months)**	36 (1.2–82.6)	48 (32.7–63.6)	<0.001	47 (26.3–58.2)

Acronyms pT1, pT2, pT3, pT4; pN0, pN1, pN2, pN3; PL0, PL1, PL2, AJCC, NS, and OS are described on Table 1 legend.

## Data Availability

The data presented in this study are availability on http://hdl.handle.net/10366/149125 (accessed on 4 April 2022).

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
