# Peer review of "Prognostic Impact of EGFR Amplification and Visceral Pleural Invasion in Early Stage Pulmonary Squamous Cell Carcinomas Patients after Surgical Resection of Primary Tumor"

_cancers, 2022, doi:10.3390/cancers14092174_

Round 1

Reviewer 1 Report

  1. It would be good  to include a comment regarding the high percentage of males in your patient cohort in your  discussion.
  2. It would be good to include   comments regarding the low % of EGFR amplifications and of EGFR amplifications and visceral pleural invasion in your discussion. 
  3. If your observations regarding EGFR amplification are confirmed, do you believe that they have therapeutic implications for all patients with squamous cell lung cancer, for patients with completely resected tumors and  no lymph node metastases, for stage IV patients?  
  4. Was EGFR expression by immunohistochemistry (1+, 2+, #+) (H score < 200 vs >200) included in you multivariate analyses?
  5. You have stated that evaluation of EGFR amplification in a larger number of patients is needed in your abstract,  it would be good to include a similar comment at the end of your discussion. 

Author Response

Comment 1.- It would be good to include a comment regarding the high percentage of males in your patient cohort in your discussion.

Answer to comment 1.- Following the comment of the reviewer a new paragraph has been added to the end discussion section of the revised version of the manuscript, discussing the high percentage of males in the series of patients analyzed. References have been also updated.

Comment 2.- It would be good to include comments regarding the low % of EGFR amplifications and of EGFR amplifications and visceral pleural invasion in your discussion.

 Answer to comment 2.- According to the suggestion of the reviewer a new paragraph has been added in the discussion section of the revised manuscript about the % of EGFR amplifications and of EGFR amplifications and visceral pleural invasion detected in our patient cohort.  

Comment 3.- If your observations regarding EGFR amplification are confirmed, do you believe that they have therapeutic implications for all patients with squamous cell lung cancer, for patients with completely resected tumors and  no lymph node metastases, for stage IV patients? 

 Answer to comment 3.- We believe that all patients with EGFR amplification will benefit from treatment, regardless of the stage and histologic type of the tumor, as occurs in other neoplasia, such as gastric cancer. Additional information on this has been added in the discussion section of the new version of the revised manuscript. References have been also updated. Comment 4.- Was EGFR expression by immunohistochemistry (1+, 2+, #+) (H score < 200 vs >200) included in you multivariate analyses? 

Answer to comment 4.- In the multivariate analysis, EGFR expression by immunohistochemistry was added but did not reach statistical significance. This information has been included in the results section (3.4. EGFR protein expression and copy number) in the new version of the revised manuscript

Comment 5.- You have stated that evaluation of EGFR amplification in a larger number of patients is needed in your abstract, it would be good to include a similar comment at the end of your discussion.

 Answer to comment 5.- A new sentence have been added at the end of the discussion of the revised manuscript, in which the evaluation of EGFR amplification in a larger number of patients is needed is specifically highlighted.  

Reviewer 2 Report

The manuscript entitled “Prognostic impact of EGFR amplification and visceral pleural invasion in early-stage pulmonary squamous cell carcinomas patients after surgical resection of primary tumor” submitted by Chinchilla-Tábora et al. is interesting. 
However, there are some aspects to be clarified:
1.    The names of the authors listed on the manuscript (p. 1) are incoherent (e.g. “LM. Chinchilla-Tábon MD”, “José María Sayagués PhD”, “Ludena MD, PhD”) and should be homogenized (first name(s), surname(s), title(s)).
2.    The “citations” section (p. 1) is incoherent (i.e. “MD, C.-T.; PhD, J.M.S.; Morais, G.; MD, M.R.; PhD1; MD, L.; PhD1”). This should be corrected (i.e. surname(s), initial(s), surname(s), initial,…).
3.    The citations in the bibliography (p. 13-15) are inconsistent with incomplete/incorrect author abbreviations and/or unnecessary additionally hyperlinks (e.g. “WD T, E B, AP B, A M, AG N. WHO Classification of Tumours of the Lung, Pleura, Thymus and Heart [Internet]. [cited 2022 Feb 28]. Available from: https://publications.iarc.fr/Book-And-Report-Series/Who-Classification-Of-Tumours/WHO-Classification-Of-Tumours-Of-The-Lung-Pleura-Thymus-And-Heart-2015”, “Fr H, M V-G, Pa B, Wa F, R D, N T, et al. Molecular predictors of outcome with gefitinib in a phase III placebo-controlled study in advanced non-small-cell lung cancer. J Clin Oncol Off J Am Soc Clin Oncol [Internet]. 2006 Jan 11 [cited 2022 Feb 28];24(31). Available from: https://pubmed.ncbi.nlm.nih.gov/17075123/”, “Zhang X, Zhang Y, Tang H, He J. EGFR gene copy number as a predictive/biomarker for patients with non-small-cell lung cancer receiving tyrosine kinase inhibitor treatment: a systematic review and meta-analysis. J Investig Med. 2017 Jan 1;65(1):72–81.”). This should be homogenized according to the journal guidelines.
4.    What do you mean with “Lung cancer is currently the most prevalent malignant neoplasm worldwide” (p. 1). Please specify “currently”.
5.    You write “with a high rates of morbimortality” (p. 1). Is this singular or plural? Please correct.
6.    You write “T-primary tumor stage” (p. 1). Do you mean pT-stage according to the UICC? Please specify.
7.    Abbreviations should be explained by first use and then used. You write “SCC” in line 25 (p. 1) and explain it “squamous cell carcinoma (SCC)” in line 28 (p. 1). Please correct.
8.    Your write “and cores of normal tissue were used as controls” (p. 3). These were from the same patients as the tumors or from other patients? Please specify.
9.    How did you define the cut off for copy number amplifications (p. 4, p. 11)? There are no details on this and there is no reference. Please specify.
10.    How did you define IHC protein overexpression (p. 11). There is no reference. Please specify.
11.    You write “two experienced pathologists interpreted the results and were fully concurring with our observations” (p. 4). Can you give more details on this? Who were these two pathologists (reference to author list)? This can not be found in the “author contributions” section.
12.    You write “we show that the presence of high levels of 7p12 (EGFR) amplification” (p. 12). What does “high levels” mean in mathematical/statistical terms? Please specify and give an adequate reference.

Author Response

REVIEWER 2:

Comment 1.- The manuscript entitled “Prognostic impact of EGFR amplification and visceral pleural invasion in early-stage pulmonary squamous cell carcinomas patients after surgical resection of primary tumor” submitted by Chinchilla-Tábora et al. is interesting.

However, there are some aspects to be clarified:

  1. The names of the authors listed on the manuscript (p. 1) are incoherent (e.g. “LM. Chinchilla-Tábon MD”, “José María Sayagués PhD”, “Ludena MD, PhD”) and should be homogenized (first name(s), surname(s), title(s)).

Answer to comment 1:  We thank the reviewer for the overall positive comments about the paper and we also appreciate the reviewer for pointing out this mistake that has now been corrected.

Comment 2.- The “citations” section (p. 1) is incoherent (i.e. “MD, C.-T.; PhD, J.M.S.; Morais, G.; MD, M.R.; PhD1; MD, L.; PhD1”). This should be corrected (i.e. surname(s), initial(s), surname(s), initial,…).

Answer to comment 2: The “citations” of the manuscript have been carefully revised and the misspellings corrected.

Comment 3.- The citations in the bibliography (p. 13-15) are inconsistent with incomplete/incorrect author abbreviations and/or unnecessary additionally hyperlinks (e.g. “WD T, E B, AP B, A M, AG N. WHO Classification of Tumours of the Lung, Pleura, Thymus and Heart [Internet]. [cited 2022 Feb 28]. Available from: https://publications.iarc.fr/Book-And-Report-Series/Who-Classification-Of-Tumours/WHO-Classification-Of-Tumours-Of-The-Lung-Pleura-Thymus-And-Heart-2015”, “Fr H, M V-G, Pa B, Wa F, R D, N T, et al. Molecular predictors of outcome with gefitinib in a phase III placebo-controlled study in advanced non-small-cell lung cancer. J Clin Oncol Off J Am Soc Clin Oncol [Internet]. 2006 Jan 11 [cited 2022 Feb 28];24(31). Available from: https://pubmed.ncbi.nlm.nih.gov/17075123/”, “Zhang X, Zhang Y, Tang H, He J. EGFR gene copy number as a predictive/biomarker for patients with non-small-cell lung cancer receiving tyrosine kinase inhibitor treatment: a systematic review and meta-analysis. J Investig Med. 2017 Jan 1;65(1):72–81.”). This should be homogenized according to the journal guidelines.

Answer to comment 3: As we have commented above, all citations of the manuscript have been carefully revised and the misspellings corrected, according to the journal guidelines.

Comment 4.- What do you mean with “Lung cancer is currently the most prevalent malignant neoplasm worldwide” (p. 1). Please specify “currently”.

Answer to comment 4:  We thank the reviewer for noting this mistake that has now been corrected.

Comment 5.- You write “with a high rates of morbimortality” (p. 1). Is this singular or plural? Please correct.

Answer to comment 5: We thank the reviewer for noting this mistake that has now been corrected.

Comment 6.- You write “T-primary tumor stage” (p. 1). Do you mean pT-stage according to the UICC? Please specify.

Answer to comment 6: Following the comment of the reviewer, the text of the introduction section describing “T-primary tumor stage” has been corrected according to the UICC

Comment 7.- Abbreviations should be explained by first use and then used. You write “SCC” in line 25 (p. 1) and explain it “squamous cell carcinoma (SCC)” in line 28 (p. 1). Please correct.

Answer to comment 7:  We have reviewed in detail the abbreviations of the manuscript and fixed the errors identified.

Comment 8.- Your write “and cores of normal tissue were used as controls” (p. 3). These were from the same patients as the tumors or from other patients? Please specify.

Answer to comment 8: The sentence has been rewritten to clarify that the controls correspond to paired samples of the patients studied.

Comment 9.- How did you define the cut off for copy number amplifications (p. 4, p. 11)? There are no details on this and there is no reference. Please specify.

Answer to comment 9: Following the comment of the reviewer, now it is defined the cut off for copy number amplifications detected, in the materials and methods section (2.4. FISH analysis) of the new revised version of the manuscript.

Comment 10.- How did you define IHC protein overexpression (p. 11). There is no reference. Please specify.

Answer to comment 10: A new paragraph has been added in the material and method section (2.3. Immunohistochemistry) of the revised manuscript about the IHC expression criteria used.

Comment 11.- You write “two experienced pathologists interpreted the results and were fully concurring with our observations” (p. 4). Can you give more details on this? Who were these two pathologists (reference to author list)? This can not be found in the “author contributions”

Answer to comment 11: We thank the reviewer for noting this important mistake that has now been corrected. “with our observations” has been deleted. The two pathologists are author of the present study (LMCH and MDL), information that has also been included in the author contributions section.

Comment 12.- You write “we show that the presence of high levels of 7p12 (EGFR) amplification” (p. 12). What does “high levels” mean in mathematical/statistical terms? Please specify and give an adequate reference.

Answer to comment 12: As we have previously commented, the cut-off for detected copy number amplifications is now included in the materials and methods section (2.4. FISH analysis) of the new revised version of the manuscript, including the term "high levels", when the amplification was found in the form clusters of hybridization signals (Figure 2).

Reviewer 3 Report

Dr Chinchilla-Tabora and Coll reported the incidence and prognostic role of EGFR 7p12 amplification in a cohort of patients with squamous cell carcinoma of the lung.

The paper is interesting, well-written and organised.

Two aspects can be improved before considering the paper for publication:

  1. the logic of the research is linear, a part from the section in which the score is developed. Why authors selected first T descriptor and then they moved to pleural invasion, which is incorporated into T2 definition?
  2. SCC is the type of NSCLC more related to active smoking. It would be very interesting compare EGFR amplification between active and former smokers, if this information is available

Author Response

REVIEWER 3:

Comment 1.- The paper is interesting,well-written and organised.

Two aspects can be improved before considering the paper for publication:

  1. the logic of the research is linear, a part from the section in which the score is developed. Why authors selected first T descriptor and then they moved to pleural invasion, which is incorporated into T2 definition?

Answer to comment 1: We thank the reviewer for his positive comments about the manuscript and the work contained in it.

We are aware of this, however we analyzed the variables independently (separately), both being significant in the multivariate analysis.

Comment 2.- SCC is the type of NSCLC more related to active smoking. It would be very interesting compare EGFR amplification between active and former smokers, if this information is available

Answer to comment 2: Unfortunately, we only have this available information on 49 patients, of which 46 (94% of cases) were related to smoking. The four cases with EGFR amplification detected in this cohort of patients were found in the group of active smokers. This information is now included in the results (3.3. Clinical, histopathological, and biological disease characteristics of SCC patients according to the number of copies of EGFR gene detected by FISH techniques) section of the revised manuscript.
